# Access to Higher Education in French Africa South of the Sahara

Catherine Coquery-Vidrovitch 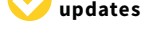

Department of History, University of Paris, CEDEX 5, 75231 Paris, France; catherine.vidrovitch@orange.fr

**Abstract:** This article examines the evolution of the educational situation in French West Africa (FWA) and French Equatorial Africa (FEA) from the onset of colonization until independence. Our central theme is the tragic deprivation endured by the public school system, especially in FEA, which handed over primary education to Catholic missions and slowed down secondary education; in FWA, only one university was belatedly created in Senegal (1958). The education of girls remained non-existent. The article is based upon a large number of mostly unpublished doctoral works, a handful of published studies, and half a century of personal inquiries, conducted mainly in Gabon, Congo and Senegal. This paper establishes a connection between the lack of political skills based upon Western standards of the colonized peoples on the eve of independence to the training of their civil servants which was drastically limited to secondary school education and the major hurdles involved in obtaining French nationality except for the residents of the Four Communes of Senegal. At the time of independence, only a few thousand colonized people had reached the level of university that was being established in the early 1950s; even fewer received scholarships to study in France. This shortage of trained personnel in administration and education required massive recourse to French "coopérants", whose presence would only gradually diminish from the 1970s.

**Keywords:** FWA (French West Africa); FEA (French Equatorial Africa); Senegal; Dahomey; coopérant (French development worker); originaire; indigénat (native code); religious mission; high school; évolué

## 1. The Educational Deficit

This article examines the evolution of the educational situation in French West Africa (FWA) and French Equatorial Africa (FEA) from the onset of colonization until independence. Our central theme is the tragic deprivation endured by the public school system, especially in FEA, which handed over primary education to Catholic missions and slowed down secondary education; in FWA only one university was belatedly created in Senegal (1958). Girls' education remained nonexistent.

The article is based upon a handful of published studies cited in the bibliography, a dozen of mostly unpublished doctoral dissertations ("theses de 3e cycle"), most of them written by Gabonese students and which may be found mainly at the Libreville university but also in various French universities, and also my own work (included in the bibliography) (Assa Mboulou 2003; Metegue 1974; Ndoume 2010; Nguema Ango 2010; Essone 2006; Mboumba 1972). They are based upon archival materials, found either in Africa or in France, dating back to a time when the colonial administration was attentive to keeping reports on schools. It also resulted from half a century of personal inquiries conducted mainly in Gabon, the Congo, Burkina Faso and Senegal. My research was carried out in the Senegalese national archives, or, in the case of equatorial Africa, at the ANOM (the National Overseas Archives center located in Aix-en-Provence), where I gleaned material without focusing specifically on the theme of education. My work amounts above all to a patient work of observation made possible by some 50 years of repeated stays in French-speaking Africa. I have made a number of visits to schools and classrooms, and I have met with many teachers at all levels of education, from schoolmasters to ministers. I have gathered material over a period of time long enough to assess the evolution of

both school and politics on the issue of education. The question has been renewed by the publication of two major doctorates devoted to the training of French-speaking teachers in Africa south of the Sahara (Barthélémy 2010; Jézequel 2002).

Of course, schooling was entirely in the hands of the colonial powers. A notable difference between English and French schooling is the French preference for public rather than private and missionary schooling. The 1905, French law barring the State from financing the Church had a major influence, all the more so that in francophone areas, French was imposed everywhere as the language of education, including in primary and missionary schools. This may help explain why schooling had a slow start and development.

In French Africa, the schooling of colonial African personnel was uneven and weak. The presence of French teachers was justified by colonizers by the lack of local workers with adequate skills, i.e., Western skills. This was obvious in French Equatorial Africa (FEA). Today, the absolutely pitiful state of "modern" education in colonial and neocolonial French-speaking Africa is difficult to imagine, yet the situation was specifically linked to the near-total absence of Africans able to gain French citizenship rights. Most Africans were "natives", and natives had no rights. Their only hope to obtain French rights (liberty, equality, fraternity) was to obtain French citizenship, and this was extremely unusual. It may be difficult today to understand that the main hope of most Africans, as late as the 1950s, was not yet to gain independence; it was to gain French citizenship. They all knew a successful example, the exceptional French citizenship obtained in 1916 by the inhabitants borne in four "communes" of Senegal: Gorée, Saint Louis, Rufisque and Dakar[1]. It was not to be obtained elsewhere before 1956.

It is also necessary to distinguish between FEA—which was abandoned until the 1950s, almost like the Belgian Congo, where colonial authorities overlooked the necessity for higher education of the "natives"—and French West Africa. The latter benefited from the relative progress of the colony of Saint-Louis du Senegal, an island occupied since the mid-17th century, thanks to the citizenship granted in 1916 to the natives of the four communes and to the only male teacher training school in the two federations, the École William Ponty. The very small number of "citizens" or those considered as citizens makes it possible to understand the tragic dearth of African political personnel at the time of independence. Those who achieved a schooling level enabling them to go to university did so late because there were only a few dozen students from FEA and FWA in French universities in the 1930s—a relative boom that materialized only after 1945—and really applied only to West Africa. This explains why studies of the system have focused mainly on the elementary level of schooling (that mostly missionaries were responsible for) which was the only one completed by almost all the lower and middle-level African civil servants. Several quality studies were conducted on the subject, most often in the form of Ph.D. dissertations written by African and French doctoral students, but unfortunately most of them remain unpublished, except for a few serious studies (Bouche 1975; Reynaud-Paligot 2021). Biographies were written on a few prominent figures: André Matswa (Gondola 2021), Blaise Diagne (Duquenet 1977; Dieng 1990), Lamine Gueye (Dieng 2013). The only researchers who have taken a direct interest in the complex relationship between (French) nationality and political citizenship have done so especially for the most recent period, after the Second World War (Coquery-Vidrovitch 2001; Cooper 2014). The small number of those who were "French citizens" until then (except for the "originaires", Africans borne in the Four Communes of Senegal: Johnson 1971; Diouf 2013) accounts for the very limited interest taken by researchers in the question. However, it is essential to understand the reasons for the lack of skills in the first twenty years of independence, essentially due to the extreme shortage of middle managers and assistants whose education was reduced to the bare minimum.

## 2. The Situation in FEA

In 1960, all of the rare French Equatorial African graduates had gone to confessional schools. The missionaries had begun their work in Gabon from the 1840s. The American Board of Commissioners for Foreign Missions (ABCFM) established a mission and a school in Baraka, Gabon in 1842 (Bucher 1973, pp. 380–81). The French seized Libreville in 1843 in the form of a trading post which was later to give birth to the city (Lasserre 1958). Catholic missionaries followed in 1844. Elsewhere, colonial settlement did not begin until the 1880s (1885 in Brazzaville and 1920 only in Chad, until then military territory).

Before the early 20th century, elementary education was minimal. It was placed exclusively in the hands of missionaries. Public schools appeared tentatively at the initiative of a few soldiers of good will—as noted by a Brazza companion during his 1905 inspection mission in Equatorial Africa, more precisely in Brazzaville[2]. When the administration took over responsibility for education in 1907, the situation hardly changed. Indeed, the main goal of the French colonial educational policy in Gabon, as everywhere else, was to maintain the colonial system in place. Like its predecessor, the goal of secular education was to make the colonized assimilate to the basis of Western culture. The programs were therefore geared toward achieving the acculturation of the indigenous people (Nguema Ango 2010).

Beginning in 1905, the law separating church and state made it difficult for public authorities to finance missionary schools in the colonies, as it did in France. For a long time, schooling was limited to the elementary level; the only relative exception was Cameroon, a former German colony taken over by France after World War I, which had benefited from the pre-war German missionary effort.

In 1914 in Cameroon, 40,000 students attended German schools, replaced after 1918 by Presbyterians and French Catholics. It was the only territory in Equatorial Africa to have established in 1937 a Superior School (for education past the certificate of primary studies) in Yaoundé. Until 1923 and all over French Equatorial Africa, everything was in the hands of missionaries, from the syllabi to the length of studies, without any public control. In 1884, the first elementary school opened in Libreville in 1845, (when French colonizers settled there) by Father Bessieux, was established as a secondary school at St. Peter's mission; in 1895, the establishment of the Latinists created in 1856 was transformed into a secondary school seminary. Between 1923 and 1945, however, the rivalry between the Mission (essentially the Fathers of the Holy Spirit) and the colonial administration became obvious. The latter, for lack of means, had little choice but to leave educational policy to the missionaries. Both the missionaries and the administration gave priority to vocational education to the detriment of general education, giving pride of place to manual work in the students' weekly schedule. The colonizing power and the missionaries were both supporters of the colonial system and made sure that they would collaborate to back up the administration and evangelization efforts. For girls, things were even simpler. They were taught their catechism, some reading and writing, and above all cooking, laundry and sewing. The language of the colonizers was imposed as the language of education in 1922 (1914 in FWA)[3], starting in the first years of school. The certificate of primary studies was made accessible to boys from 1930 only. Few children ever reached that level:

> "Too many pupils each year leave our schools with little education—they are barely literate, with some writing skills, they have memorized a number of French words the exact meaning of which they sometimes do not know; in short, they are educated just enough to stray away from the land and despise their brothers who have remained in the village, but unable to use the simulacrum of an education they are so childishly proud of to make a living. None of them is able to become a writer, a typist, an accountant. More often than not, they are downgraded, disgruntled beings, mere parasites of the working community."[4]

(Guth 1990, p. 75)

The first secondary school in FEA, the École Édouard Renard, was founded in 1935 in Brazzaville. After receiving their certificate of primary studies, students had two years to prepare for its competitive examinations in regional schools. Schooling was to last for three years. It comprised several sections, all designed to train those who would perform auxiliary work—teaching assistants, accountants (senior staff of native administrative agents), health assistants, agricultural instructors, etc.—and it also included vocational sections: iron and wood workshops, bookbinding as well as leather work. A model farm crowned it all.

Secondary education remained nonexistent, further delaying the proper education of a modern elite (Perlstein 1943, pp. 130–35; Bouche 1975; Keese 2007), as colonizers called it for Westernized people; the condition was to earn a diploma, most of the time the "certificat d'études", or at least to prove that one had adopted a European lifestyle because the official purpose of French colonialism was assimilation. It was not until 1947 that the first public secondary school was opened in Gabon. The first missionary secondary school opened in 1948; it was the Collège Bessieux. Cameroon was the only country with 200 pupils or scholarship students studying in France. After the war, the French FIDES program (Investment Fund for Economic and Social Development) opened a few technical schools, and 34 students for the entire FEA received scholarships to study in France in the late 1940s (Bernault 2020). In other words, when independence was achieved, apart from a few privileged individuals who had been able to apply to the William Ponty School of West Africa, there were few middle managers and even fewer senior managers in FEA. The tragedy of this federation was the lack of training of an "elite", due to the colonizers' conviction that the primitive level of the local populations made it almost impossible: "Why these differences among our various colonies? . . . They are due to the degree of mental development of the races and to their possibilities of assimilation and adaptation". (Perlstein 1943, p. 130).

Few individuals were able to benefit from a genuinely thorough education. Both the missionaries and the administrators had been content with teaching reading, writing and manual skills. Such attitude of the missionaries and the colonizing power is responsible for delaying the education of an intellectual elite worthy of that name, in FEA as in other African colonies. In 1929, the number of school-age children in FEA was 400,000 but there were only 4000 students in public schools; the following year, in 1930, only 23 students for all of FEA passed the certificate of primary studies. Three years later 39 students did it (Bernault 2020). The educational situation in Gabon was not as poor as it was in other colonies within Equatorial Africa such as Chad, Oubangui-Chari and the Middle Congo. The reason for this is to be found in its scarce population (probably amounting to little more than 400,000 at that time). By 1950, primary education had improved: there was schooling for 70% of the children in Gabon and Congo, but the education rate remained very low in Oubangui-Chari (27%) and even lower in Chad (8%) (Bernault 2020). At the end of the primary school cycle, the certificate of studies guaranteed the securing of a position within the local administration. Beyond that stage, education was limited to a bare minimum. In 1940, the École Supérieure Édouard Renard in Brazzaville was the only serious educational institution in French Equatorial Africa.

The administration needed those "auxiliaries", many of whom were able to make a name for themselves. Léon Mba was one of them. He was to become the first president of the newly independent Gabon (1961–1967). He received his certificate of studies in 1909 from the seminary. He entered the colonial administration as a customs clerk and became the chief of an administrative district. His reputation as a jurist comes from his work transcribing the customs of the Fang people (Mba 1938), which he produced during his exile in Oubangui-Chari. His activism and intelligence worried the administration. Penniless and exiled in the 1920s, he was lucky to be recruited as a "boy" by an open-minded colonial administrator, Gaston Guibet, who encouraged him in this path (Guibet 1966)[5]. He was one of those exceptional individuals who could be counted on the fingers of one hand.

He was not the only one in FEA to become a member of the "elite". The region, placed under the leadership of free France in 1940, understood that it needed native executives. However, the latter suffered from a menial status because they were subjected to the indigénat (native code), putting them under the authority of the local administrator in a binding manner because the latter concentrated executive and judicial powers: he had the power to inflict financial penalties and prison terms for a whole series of offenses (such as tax evasion, insolence, etc.). A mere trickle of people obtained French citizenship, essentially in FWA (hardly more than 30 per year after the 1930s) (Coquery-Vidrovitch 2001), after an investigation guaranteeing that the "native" had a sound general level of education and had adopted a European lifestyle. Free France thus invented an intermediate status in FEA on 29 July 1942, namely that of 'notable évolué' (educated native), which exempted them from the penalties associated with the indigénat while they remained ruled by customary provisions and indigenous courts (Perlstein 1943). After the Second World War, they were able to participate in electoral colleges for local assemblies. In short, this marked the statutory invention of the "native elite", stemming from the first efforts at providing an education likely to make good colonial "auxiliaries".

The boys who made a name for themselves may be counted on the fingers of one hand, and the public school system did not yet accept girls who attended missionary schools. A few gifted young men who were chosen by their teachers were admitted to the William Ponty School in Senegal. After World War II, others were able to begin their university education in Bordeaux or Paris; however, they did not break through before independence. One of the very first ones to stand out was Joseph Ambouroué-Avaro (1934–1978). He had completed primary education in the 1940s at the MontFort school in Libreville. After his certificate of studies, he received his degree with a scientific major from Bessieux high school and wanted to study medicine. Such educational progress was supported by the position of his father, who was one of the "barons" of President Léon Mba's regime. Eventually, on the advice of one of his uncles, he opted for a career in the humanities, undertaking the preservation of the collective memory of the country. Thanks to a scholarship to attend the Sorbonne, he obtained his graduate degree in Political Science. Because he was committed to the ideals of democracy, development and justice, his activism with the General Association of Students of Gabon (AGEG) and the Federation of Students of Black Africa in France (FEANF) cost him his scholarship. He nevertheless managed to defend his dissertation in 1969, supporting himself by teaching history and geography in the high schools of Chauny and Nogent-Le-Rotrou (Ambouroue-Avaro 1981 [6]; Biography 2009). When he returned to his native country that same year, he was the first Gabonese university historian. His political opposition to President Bongo's regime, his criticism of the calamitous management of the country, the violations of democracy and the social injustice that plagued Gabonese society, placed him under constant threat. Had he not died prematurely in an accident that was probably a political assassination, he would have had a great career.

In the Congo, Théophile Obenga was born in 1936 and received a scholarship to study in Bordeaux after the war. He was to become a disciple of the linguist Cheik Anta Diop and held several ministerial positions in "Marxist-Leninist" Congo after 1963. Henri Lopes, a Congolese historian and novelist born in 1937, owed his political career to the fact that his mother lived with a white man who adopted him. He attended school in Brazzaville and Bangui, and then went to Paris and Nantes for higher education in 1949. He completed his studies at the Sorbonne in 1963 (Lopez 2018)[7]. He then returned to teach in Brazzaville but soon embarked on a brilliant political career. In the various governments of the "Marxist-Leninist" period, he held important cabinet positions, among which the ministry of the economy; I met him when he was Prime Minister (between 1973 and 1975). He ended his career as Congo's ambassador to France. His case is quite exceptional (Lopez 2018). The few Parisian students from FEA, like their fellow FWA students, belonged to the militant core of the Federation of black African Students in France (Fédération des Étudiants d'Afrique noire en France) after the war.

With the exception of these few great figures, and a limited number of higher education graduates who began training, in France or in their native country, only from the 1980s onward, qualified personnel remained extremely limited. There had also been far fewer soldiers recruited for the First World War in FEA than in FWA (about 10,000 as opposed to nearly 200,000: Michel 1982). These "tirailleurs" (riflemen), who were not subjected to the indigénat when they returned from the front, had discovered another way of living and thinking in France; no longer considered as strictly natives but not yet citizens, they benefited from a kind of aura. Pierre André Grenard Matsua was one of them and became famous.

He was one of the first politicians of the country to be eliminated by colonial power: Matswa (1899–1942) was indeed inspired by the success of the "originaires from the four communes" of Senegal who were rewarded with French citizenship during the First World War. An active militant in France, in the Congo he actually suffered from being a trailblazer due to the absence of an elite likely to prevail over a very repressive colonial power. After a Catholic education with the Fathers of the Holy Spirit, he became a catechist in the Pool region; eager to improve himself, he gave up the apostolate to go to Brazzaville and became interested in the relations between white and black people in the colonies and in the future of his region, the Congo. He then began to make a name for himself through his knowledge of the sociopolitical problems of the Pool region. As he dreamed of going to France, he finally obtained a temporary pass to go to Marseilles in 1923 and joined the Senegalese "Tirailleurs" in 1925 to take part in the Rif War. In 1926, he settled in Paris as an accountant at the Laennec Hospital and worked his way through a political education, attending evening classes for "natives".

Associating with other emigrants in left-wing circles led him to discover new ideals to sustain the struggle against the injustice and harassment of colonization. Under the name of Grénard, he founded the Association of Natives from French Equatorial Africa (Amicale des Originaires de l'Afrique Equatoriale Française) in Paris in 1926, a name which was obviously inspired by the existence of the "originaires" of Senegal. As any political activity was closely monitored, the association was meant to provide aid for educational purposes to "help blacks released from military service in France", thereby officially rejecting any political stance. The objective was to train a Congolese elite and help them achieve the independence of their country by peaceful means. Emissaries were sent to the Congo, they were arrested, and Matswa himself was wrongly accused of financial misappropriation of membership fees in the Association; he was sentenced in 1930 to three years in prison and banned from residing there for ten years. He escaped and returned to France to enlist in the "phony war". After being wounded, he was arrested again in April 1940 and sent back to the Congo to be sentenced to life imprisonment. He died in suspicious conditions in 1942. In the meantime, his arrest had caused major unrest in the Brazzaville region. However, his mysterious death (which has remained so since the administration concealed his body) was due to influence the direction of the movement: the Congolese members of the Association, who cannot be said to have belonged to the rare educated elite, were to initiate the birth of a prophetic cult which, under the name of Matswanism, was to be used politically by the Lari, an ethnic group from his region. They were used at independence to bolster the accession to power of a defrocked former priest, Fulbert Youlou. Matswa is a fine model showcasing the wish for intellectual and political progress on the part of a colonized people confronted to the colonial rejection of promotion through education (Gondola 2021; Bernault 1996).

When I first went to Equatorial Africa in 1965, five years after independence, I was struck by the lack of skills at almost all levels. Workers in general were full of good will, but obviously ignorant. There was a huge shortage of intermediate-level staff, with top level agents having at best barely achieved the certificate of studies. Hence, as I have already pointed it out, at the beginning of their existence as independent nations, the latter were plagued with a dramatic shortage in personnel at all administrative and political levels. This lack of local skills made it possible to understand, at all levels, the reasons for

the omnipresence of French "coopérants" in all fields; in education of course, but also in the administrative and technical realms, as well as in research. Due to a lack of trained personnel, the country had remained under tutelage. The highest diploma most young Africans dreamed of was the certificate of primary studies. The situation only began to change after the 1990s, when a new and better-trained generation began their education.

### 3. French West Africa, the "Jewel" of French Black Africa?

The FWA also benefited from collaboration schemes. However, its legacy was vastly different; as early as the 1920s, the FWA sent a few privileged scholarship recipients to pursue their training as teachers at the Ecole Normale (teacher training school) in Aix-en-Provence. During the colonial era and almost until independence, the great training institution for Africans was the William Ponty School. Except for a few "originaires" who took a different path, the greatest achievement for boys was to become a teacher or an assistant to a physician. Nevertheless, compared to FEA, FWA seems to have been extraordinarily privileged. How surprising indeed!

Saint Louis had been occupied by the French since the middle of the 17th century. A culturally assimilated "Creole" bourgeoisie developed there (including at the time of the slave trade) (Pasquier 1987). In 1884, the "Special Secondary Education" section was created there and run by the friars of the Plöermel Institute, which included secondary education, technical education and a business course (four years of practical studies). Taken over by the State in 1893 in the form of a Higher Primary School (Faidherbe School), in 1910 (as a consequence of the law separating Church and State) it became an establishment providing the same education as the secondary and high schools of France, a vocational school and a teacher training college. The latter was transferred to Gorée in 1913 under the name "Gorée Teacher Training School". In 1915, it was renamed the "William Ponty teacher training school" in honor of the governor general who had just died in Dakar. In 1921, the school was reorganized and integrated into the Professional School of Administrative and Commercial Apprenticeship of the former "Faidherbe School". It took the name of "William Ponty School" for short, with three different pathways: a section for training teachers, a general section (for administrative and commercial agents) and a section preparing candidates for the FWA School of Medicine in Dakar. The administrative section was closed in 1924.

In Saint-Louis, the former Faidherbe School became the Lycée Faidherbe (Faidherbe high school) in 1924[8]. The citizenship granted to residents of the four communes since 1916 allowed hand-picked African students to benefit from French education, which was otherwise reserved for settlers and a few privileged Creole families. The number of students remained limited: 100 at the beginning and 800 in 1950, most of them white. High level graduates from before the Second World War were rare, just a few dozen. In 1920, a secondary school was opened in Dakar; in 1936, it became the Lycée Van Vollenhoven (now the Lycée Lamine-Guèye). Most gifted young Africans attended the William Ponty School, which channeled them by receiving the brightest minds of West Africa, trained as teachers or as "auxiliary doctors" (such would be the case for Félix Houphouët-Boigny, the future president of the Ivory Coast) (Ly 2009; Jézequel 2002)[9]. Exceptions were very rare: the finest specimen was Léopold Sedar Senghor, born in 1906, who was not originally from one of the four communes, but a gifted child who was singled out by a teacher in the Saint Joseph de Ngasobil mission where he was sent until 1922, and from there to the Liberman seminary college in Dakar. Then, having been expelled from the seminary, he entered the secondary school of rue Vincens, the future Van Vollenhoven high school in Dakar. He was to preside over the West African Students' Association created in Paris in 1933. He was the first holder of an agrégation (high-level competitive exam for teachers) in Africa in 1935 and worked in France until the Second World War.

Contrary to Senghor, Blaise Diagne (1872–1934) was born in Gorée and stands out as an example of the extraordinary advantage that having French nationality represented for a native at the time. Born to a Lebou father, an ethnic group from Senegal composed

mainly of poor fishermen and peasants, he was taken under the protection of a mixed-race family who were public figures in Gorée. He proved to be a brilliant pupil at the Saint Louis elementary school (1884) and obtained a scholarship to continue his studies in Aix-en-Provence, but due to illness, he returned to Senegal to take the exam for customs officer in 1891. He made a career in the colonial administration and was the first African to be elected to the French National Assembly in 1914[10]. It was thanks to his parliamentary action in France that the natives were recognized as full-fledged French citizens—Diagne negotiated this recognition in exchange for the involvement in military service of young natives and the commitment to organize the recruitment of "subjects" from FWA (nearly 200,000) in the Great War. He thus achieved for the natives alone what all Africans dreamed of: full French citizenship rights, which guaranteed them "Liberty, Equality, Fraternity," a status which the "natives" of the Empire that was to become the "French Union" would only partially obtain in 1945 by becoming "imperial citizens" (Coquery-Vidrovitch 2001, pp. 285–305). Only the "Originaires" experienced a political life à la française between the two World wars because they were indeed French. Nevertheless, prejudices against skin color added to the great poverty of most families did little to promote social and political progress. There prevail few examples of social upward mobility, the most famous of which is Lamine Gueye (1891–1968). He is a good example of the influence of education on a gifted native at the time: he obtained the certificate for primary school studies in 1906 and then the Brevet élémentaire in 1907 from the Faidherbe school. Then he became a teacher, prior to obtaining a Bachelor's degree in Mathematics in France, which enabled him to return as a French teacher to the William-Ponty School in Gorée. In 1912, with a handful of friends from the cultural association named "Aurore de Saint Louis" (Dawn of Saint Louis), he founded the first political group in FWA called "Jeunesse Sénégalaise" (Senegalese Youth). He started reading law during the First World War, then became the first black jurist in French Africa (Duquenet 1977). Having returned to Senegal in 1922, he reorganized the Senegalese Socialist Party in 1935 and became Dakar's mayor in 1945. His political role was that of an eminent socialist activist until after the Second World War.

Without citizenship, there was no hope before the Second World War, except to escape from colonial Africa (Desalmand 1983). This is the case of writer and filmmaker Sembene Ousmane (1923–2007), who was too rebellious to reach the level of the certificate of studies. He earned his living in Dakar as a mechanic and mason while taking an interest in cinema. After the transfer of FWA to the Free French, he joined the Senegalese "tirailleurs" in 1942. In 1946, he embarked clandestinely on a ship bound for Marseilles where he worked as a docker for ten years. He joined the French Communist Party, read extensively from the libraries of the General Workers' Confederation and took the classes offered by the French Communist Party. In 1956 he published his first novel, *Le Docker noir* (The Black Docker); he enrolled in the Moscow Film School in 1961 and the following year he produced his first short film *Borom Sarret* (The Carter). We may safely say that his entire being was bent on resisting colonization (Gadjigo and Diop 2010; Gadjigo 2013).

Outside Senegal, education was more limited, except in Dahomey (now Benin), which was at one point nicknamed the "Latin Quarter of FWA" because it was said to send almost as many candidates to the William Ponty school as Senegal did. For other countries, there was hope of social progress only for "non-indigenous" people, that is to say overwhelmingly World War I veterans. In Dahomey, some families were privileged due to their history: such was the case of some Afro-Brazilians who had returned to Africa and enriched themselves thanks to the slave trade. Their families could afford their education, even in France. In the early 1920s, an "educated African" of this kind called Dorothée Joachim Lima launched a weekly critical newspaper in Cotonou in partnership with Jean Adjovi. The newspaper called *Le Guide du Dahomey* circulated from 1920 to 1922, with a total of 88 issues (Campbell 1998). Dorothée Lima was naturalized French, educated in a missionary school in Porto-Novo, and he entered the colonial administration before the 1914 war. A few years later, the newspaper was revived by the same men under the title *La voix du Dahomey* (1927–1957) and under the official name of Dorothée Lima, which was

supposed to constitute protection from repression. This did not prevent the governor of the colony from filing a lawsuit against the newspaper, a lawsuit that was eventually lost by the administration in 1936 under the Popular Front government (Codo 1984).

In 1920, only 5% of Senegalese children went to school (Bayet 1972, pp. 33–40). At the beginning of the 1939 school year, although this country was the most advanced in this respect, there was still only a very limited number of pupils in public schools: for a population of about 1.8 million, elementary school was a reality for only 13,594 pupils, 112 of whom were girls. The latter were taught by three teachers who were all French; the food served in the canteens had to be paid for by the parents[11]. After the Second World War, the colonial administration made efforts which were still insufficient. Education nevertheless improved quantitatively and qualitatively. Africans attended sec: ondary education in greater numbers: from 723 students in FWA between 1940 to 1945, they grew to 4174 in 1948 (Guth 1990, p. 78). Long elitist metropole style education became an objective for the children of the elite, but no thought was given to the problems induced by mass education nor to the transformation of the rural world. However, the number of scholarship recipients studying in France was to grow significantly, as several thousand scholarships were awarded between 1946 and 1960. The leaders of black student associations in France were all acquainted, and they met in 1956 on the occasion of the first Congress for black writers and artists organized in the Sorbonne by Alioune Diop, a young Senegalese literary scholar educated in Algiers. About sixty of them attended, all nationalities combined, francophone, Anglophone and African-American, as demonstrated by the group photo taken then (Swain 2006)[12]. We find these personalities in the political history of their respective countries. Such activism encouraged increasingly challenging young people living in the cities of FWA to demand participation in political emancipation, via school and student associations, youth movements and sports associations (Bancel 1999, 2009).

Conversely to the situation in FEA, where virtually none of them were able to break through such glass ceiling, in FWA a few women managed to get an education as early as the 1940s (they had been allowed to become nurses and midwives since the 1920s) (Coquery-Vidrovitch 2013). The first French-speaking African female politician, Aoua Keita, came from FWA (Keita 1975). It was thanks to her father, a former World War I tirailleur, that she was sent in 1923 to the first girls' school in Bamako, against the wishes of her mother for whom it was scandalous to send a girl to school. The father was not particularly a feminist but he was elderly and his wife had borne no sons; thus, one of her daughters had to be able to take care of her in old age. Aoua Keita always ranked among the top students, and she obtained her midwifery diploma in Dakar in 1931. Her first job was in Gao, a particularly deprived area, where she had, as "the only emancipated young girl", quite an "overwhelming success" as she put it herself (Keita 1975, p. 28). The young woman was not to be daunted and succeeded in building a maternity ward. As was to be expected, she married Diawara, a young doctor who came to work with her after graduating from the Dakar School.

Diawara was, like his wife, one of the few "assimilated" people. He was more than an "évolué"; being an "originaire", he benefited from French citizenship, and he voted in the first college (the college reserved for Europeans and the few Africans who were French citizens). Lively and well-read, the young woman was introduced to newspaper reading and politics by her husband. They joined the professional union of health workers and participated in the first major strike of the union which lasted for three months in 1945. The following year, they joined the Sudanese branch of the Rassemblement démocratique africain (African Democratic Rally). Aoua Keita began to militate by chatting with women who had given birth but did so covertly because "politics is the business of men and not yours," as the husband of a neighbor once told her. Aoua Keita entered politics because she was barren; her mother-in-law wanted that Aoua's husband to take a second spouse, which he did in spite of having been hostile to polygamy until then—therefore she divorced him in 1949. Then in middle age, Aoua Keita was an independent woman[13]; "I produced an adventurer," said her father in a surprised manner, while her mother lamented it. She

traveled alone from post to post, working at jobs which were often meant to be punishments because the administration took a dim view of her political activity made possible by a life of permanent contact with her patients and their families. She was an intelligent and passionate activist who tirelessly explained the issues at stake and rallied women to her cause (Turrittin 1993, pp. 39–99).

Aoua Keita had the ability to awaken the consciousness of people. Because she was about the only woman able to do so among an immense majority of illiterate people, she played an extraordinary role in Gao. Returning there after being away for fifteen years, she and other women managed to fight off the repression trying to influence the legislative elections of 1951 right under the nose of an overwhelmed administration. The determination of these women can be fully understood when one remembers that, in this highly Islamized region, it was inappropriate for a woman to set foot outside her home before dusk. Within a few days, on the eve of the elections, all Sudanese officials suspected of being pro-African Democratic Rally had been transferred away. Thanks to her knowledge of the law, Aoua Keita managed to obtain the voters' electoral cards and had them delivered to their homes by the women's network. Finally, on voting day, the women who were individually assisted by schoolboys in charge of reading for them, stood in front of the polling stations first thing in the morning in order to prevent fraud. The ADR won in Gao thanks to the women's vote. Aoua Keita was the first and only woman elected to the Constituent Assembly of Sudan-Mali in 1958. It took the intervention of the head of state, Modibo Keita, to make the other deputies accept her presence (Coquery-Vidrovitch 1997).

Since 1918, only one nursing school located in Dakar had been in charge of training the first Senegalese female workers. Things changed in 1938 for female teachers when a stubborn French woman obtained the right to found a teachers' training school for girls: until then, no African woman had been a teacher. At best, they had worked as auxiliaries. From then on, and on a yearly basis, a class of forty girls was trained as teachers dedicated to the French school system for the whole FWA area (Bouilly and Rillon 2016; Barthélémy 2010). Dispatched all over the federation, in spite of their limited number they played a major role, generation after generation, for girls education while they taught what they themselves had learned in their westernized school. Born to a diplomat, the novelist Mariama Ba (1929–1981) was a schoolteacher and former student of the Rufisque teacher training school (1947). She worked after independence to dismantle the inequalities between men and women in Senegalese society (Ba 1979). The daughters of the women who had attended teacher training school often chose to pursue higher education from the 1960s onward. When they entered university in 1960, there were only two girls in the sociology lecture hall of 300 students in Dakar, one of them being Fatou Sow (herself the daughter of a school teacher) who became a prominent feminist researcher (Locoh and Puech 2008)[14]. These girls—not more than a hundred at most—made up the first generation of prominent female characters who blazed trails in the public sphere (Faladé 2020). When I arrived in 1967, except for a few exceptions—there was a female anesthesiologist in the hospital, an assistant and the secretary for the history department at the university—Senegalese women did not yet have a serious place in the world of salaried work (except for a few Casamance women with a Catholic background). They appeared in work places (as secretaries, doctors, teachers, lawyers, civil servants . . . ) in the years 1975–80 at the earliest. The girls of the following generation, from the 1990s onward, began to enter political life.

## 4. Conclusions

In a nutshell, the education of executives was very limited on the eve of independence. When I arrived in Gabon and the Congo in 1965 and in the Ivory Coast in 1967, the lack of skilled and, perhaps even more so, of intermediate personnel astounded me. What was obvious and remained so for quite a long time was the nonexistence or incompetence (real or misdirected) of lower and middle African staff and often of managers, all of them male workers. Between my arrival in sub-Saharan Africa in 1965 and the 1980s, progress was slow and uneven. This was not surprising: the French school system was deficient until

independence; then the schools were overwhelmed by the demographic boom that began in the 1950s thanks to the practice of preventive medicine (based on immunization) that developed largely after the war. Education at almost all levels, including the elementary level, was still run by French people, and they were far from being numerous enough.

The highest diploma most young Africans dreamed of was still the certificate of primary studies. As we have seen, it was not enough to fill all the positions needed for the efficient management of a young independent state, which explains an almost systematic reliance upon French civil servants or "coopérants". The presence of coopérants was justified by the lack of local expertise. Most of the coopérants were initially recruited from among the former administrators of overseas territories who had lost their raison d'être, but who were often appreciated for their knowledge of the field. The coopérants working as teachers were relatively well-known and, from 1960 onward, they were present at all levels: in elementary schools, high schools and secondary schools, and, much later, universities (the first university founded in Dakar was only gradually created between 1952 and 1958) (Goerg and Raison-Jourde 2006). However, the available pool of coopérants was employed in all fields: many occupied administrative or even governmental functions, serving as prefects, chairpersons and advisors of all kinds. In 1965, the Minister of the Interior for Gabon, who had taken over the responsibilities of the ailing president treated in France and who had come to welcome me, had a "double" in the person of a special adviser, who was none other than a former colonial civil servant. The latter explained to me that he had succeeded his father at that time and was proud of having achieved what his father had begun a generation earlier. The number of coopérants began to dwindle in the 1970s, due to a protest movement that expressed itself vigorously in Dakar from 1968 onward and called for the "Africanization of executives". They belonged to the first generation that began to benefit from the efforts made by the colonial authorities in the 1950s and especially from the schooling boom accompanying the birth of independent nations. As the education of executives in Francophone Africa was just emerging at the time, it comes as no surprise that we should witness, on the one hand, the lasting weight of French cooperation in all fields (Meimon 2007), and on the other hand, the heavy legacy that this past implies in terms of dependence.

In this field, due to the relative scarcity of graduates, at the time of decolonization the link between political function and degree strengthened, independently from the militant or protest action within student associations of this "elite" stemming from the few students educated in France. As the number of scholars was extremely reduced, an aristocracy of literacy developed (Mbembé 1985; Guth 1990, p. 93). The takeover of the state apparatus at the time of independence was directly linked to the educational background of the nationals belonging to the new States who, with few exceptions, all found themselves more or less in charge of their country. The development of the civil service initially made it possible to take in any new graduate, or even to promote him or her immediately to a management position. This monopoly of diplomas often lasted more than a decade while the literacy level of employees rose. After recruiting the elite at the secondary school and then high school level, after independence came the bachelor's degree, the master's, doctoral degree and even, after the 1980s, the state thesis. Academic distinction, a determining factor in the colonial era for obtaining citizenship, continued to be an essential factor of social differentiation and frequent political promotion.

In colonial Africa, French school has thus been the cause of major upheavals in the social, cultural and political order. Nevertheless, in spite of an official French colonial mindset, but no actual wish for assimilation, social and political evolution was slower than in most British colonies (such as the Gold Coast, Nigeria or even Kenya). Under-education resulted in a stronger deficit of trained African staff on the eve of independence. Most independent states worked hard to fill the gap and to bolster at least primary education. The result was obvious in the 1970s for nations such as Congo-Brazzaville, Guinea or Burkina-Faso. Then, most children, or boys at least, were schooled at the primary level. Unfortunately, the economic crisis of the 1980s, the dramatic structural adjustment pro-

grams imposed by the IMF, and a series of civil wars often drastically disorganized what is most important for development—education. This topic would require another article to be written.

**Funding:** This research received no external funding.

**Institutional Review Board Statement:** Not applicable.

**Informed Consent Statement:** Not applicable.

**Data Availability Statement:** Datas in ANOM Archives nationales de la France d'Outre Mer, Aix en Provence France.

**Conflicts of Interest:** The author declares no conflict of interest.

## Notes

1   I explain this «anoimaly» below, when speaking of the French West Africa case.
2   Félicien Challaye was a young philosopher recruited by Brazza to draw up a study of the state of education in the Middle Congo. His report on schools (written in Brazzaville 1905) may be found in ANOM (French National Overseas Archives), Fonds Mission Brazza 26.
3   The missionary educational policy of Gabon is known thanks to the unpublished theses of several Gabonese historians: Metegue N' Nah, Mboumba Bouassa, Mezui, Ndoume Assebe, Nguema Ango (see bibliography).
4   Governor general Antonetti (FEA). 1925. *Journal officiel de l'Afrique Équatoriale Française*, 15 mai: 281. Quoted by Guth.
5   Léon Mba called to his side Gaston Guibet (1881–1973), who had retired as a governor, when he became President. Guibet confided this detail to me when I interviewed him at my home in Paris (in May 1966) after meeting him in Libreville in October 1965.
6   He defended his PhD at the Sorbonne in 1969 and it was published posthumously; Personal testimony, (Libreville 1974).
7   See his autobiography: 2018.
8   The Faidherbe High School became Sheikh Omar Foutiyou Tall High School in 1984. In 2002, 1070 students attended it.
9   Jézéquel's thesis is a prosopographic study of the 2200 graduates from the Ponty teacher training school.
10  Since the French Revolution, the "old colonies" each elected a deputy to the French assembly (although there were several interruptions in the 19th century under the different Empires). A settler held such office for Senegal until 1900, the year when a mixed-race person was elected for the first time. Blaise Diagne succeeded him in 1914.
11  National Archives of Senegal, Dakar, beginning of the 1939–40 school year, 2G 40 74.
12  https://i0.wp.com/histoireengagee.ca/wpcontent/uploads/2016/12/lumieres_noires-.jpg (accessed on 15 August 2020). To the best of my knowledge, prosopographic analysis of French-speaking students who had completed part of their training in France before independence has not been carried out, except for the former students of the William Ponty School and the Rufisque School for Female Teachers.
13  She later remarried with Mahamane Alassane Haidara, who was elected senator to the French National Assembly in 1942 to represent the peoples of the Niger loop. He remained senator until 1959.
14  Born in 1941, she had to finish her studies in Paris because sociology was not taught in 1968 at the University of Dakar. She was tenured in 1984 as a researcher at the CNRS (French National Center for Scientific Research). Thérèse Locoh and Isabelle Puech, "Fatou Sow. Les défis d'une féministe en Afrique," Travail, genre et sociétés, Vol. 20, No. 2, 3 December 2008, pp. 5–22.

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
