# Peer review of "Access to Higher Education in French Africa South of the Sahara"

_socsci, doi:10.3390/socsci10050173_

Round 1

Reviewer 1 Report

  1. Needs to discuss applications of grounded theory as noted in qualitative research
  2. Article appears more descriptive rather than analytical and prescriptive
  3. Very Eurocentric perspective in tone and quality - e.g., could use more critical discourse about cultural imperialism and colonialism in education
  4. Could benefit from a critical review about the dialectics of decolonization and current support for education for all under UN norms, etc.
  5. Could benefit from discourse about French citizenship, Declaration of the Rights of Man and of the Citizen, 1789 - which raises addition questions about the consultancy role of Thomas Jefferson in the development of that document; as well as later events such as the success of the revolution in Haiti by enslaved Africans

Author Response

My purpose being how late was colonial education in French African colonies, the question is to explain why western mode educated Africans were so few, how brilliant these few people might be. The fault was the defiance of colonial powers who avoided as much as possible high education for « natives ». This is why I insist on the French educational system most of the time limited to primary education. Of course, schooling was entirely in the hands of the colonial powers. A notable difference between English and French schooling is the French preference for public rather than private and missionary schooling. The 1905 French law barring the State from financing the Church had a major influence, all the more so as in francophone areas French was imposed everywhere as the language of education, including in primary and missionary schools. This may help explain why schooling had a slow start and development.

Therefore, in French Africa, the schooling of colonial African personnel was uneven and weak. The presence of French teachers was justified by colonizers by the lack of local workers with adequate skills, i.e. western skills. This was obvious in French Equatorial Africa (FEA). Today, the absolutely pitiful state of “modern” education in colonial and neo-colonial French-speaking Africa is difficult to imagine, yet the situation was specifically linked to the near-total absence of Africans able to gain French citizenship rights. Most Africans were “natives”, and natives had no rights. Their only hope to get French rights (Liberty, Equality, Fraternity) was to obtain French citizenship, and this was extremely unusual. It may be difficult today to understand that the main hope of most French speaking Africans, as late as the 1950s, was not yet to get independence; it was to gain French citizenship. They all knew a successful example:  the exceptional French citizenship obtained in 1916 by the inhabitants borne in 4 “communes “ of Senegal : Gorée, Saint Louis, Rufisque and Dakar. It was not to be obtained elsewhere before 1956, 4 years before independence.

Censorship being very strict, young Africans knew nearly nothing about French revolution, and were completely ignorant of the Saint Domingue revolution and Haiti departure (as were also French citizens in France). These remained late a “taboo” for French history. This is the reason why I don’t speak of it. My aim is to analyze the French way of ruling education, which is necessary to understand the situation, except a very few African scholars who emerged only after WWII and whom of course I speak of. This is still truer for African girls, who could begin to emerge in the political sphere only with independence. For a fewof them, to go further than elementary school was nearly impossible before WWII.

Reviewer 2 Report

Globally, it is a most interesting paper, which gives a welcomed overview of (under)schooling in Francophone Africa, and its consequences on Independent countries. It is well written and compelling, biographical material give a welcomed rhythm to the paper.

The paper is mainly based on secondary literature regarding colonial era, as well as on personal insights from the 1960s.  It gives an overview of colonial schooling in Francophone colonial Africa. The paper argues that the organised underschooling of said territories, precluded the training of "Modern elites" able to govern independent countries.

I recommend to publish the paper, however, I believe some modifications should be made beforehand:

My remarks address mostly:

  • Making a more nuanced argument regarding girl schooling;
  • Defining and explaining what a "modern elite" ought to be in colonial setting, probably in the introduction part;
  • Grounding the personal insights on francophone independent countries in the literature, as the last decade witnessed a rise of historical works on independent francophone countries, which regularly address coopération and governance;
  • Finally, comparing francophone setting with neighbouring colonial territories (notably anglophone, well more schooled) would give some perspective on the main argument (notably in the conclusion).

More detailed remarks follow:

Abstract

What do you mean by “lack of political skills”? Western skills?

The education of girls did not remain "non-existent", as indicated in the paper and in the literature.

The Educational Deficit

The data corpus is not clearly indicated, it would be worth having a paragraph dedicated to sources and/or secondary literature used in the paper.

In the abstract, you indicate the following: “The paper is based upon a large number of mostly unpublished doctoral works most written by African graduate students, a handful of published studies, and half a century of personal inquiries, mainly in Gabon, Congo and Senegal.” This raises a lot of questions.

  • What material was collected during these inquiries? Interviews? Archive material? Which were specifically used for the present paper? And/or, alternatively, previously published in more details in previous books/papers?
  • On the same note, the doctoral works could be detailed. What are their status? Sources? Historiographical material? Where were they found? Which universities? How many of them ("large number")? Etc.

Review of literature: One could also refer to Barthélémy, Africaines et diplômées à l’époque coloniale (p.2).

The situation in AEF

“further delaying the education of a modern elite”: the definition of the said modern-elite could be expanded and defined (eventually in a comparative perspective with other Empires). What do you mean by “modern elite” ? See notably Alexander Keese, Living with Ambiguity. In his book, he he set out a nuanced analysis of late-colonial political elite either deemed “traditional” or “modern”.

The final paragraph (p. 6) is a bit vague. I agree with the importance of ethnography, however, it would be worth precising which administrations are referred to, as well some precise references (See for instance: "Workers in general were full of good will, but obviously ignorant."), in order to ground such affirmations. As far as "coopération" is concerned, I would notably recommend relying on Julien Meimon's work.

French West Africa, the “jewel” of French Black Africa?

Girl schooling was indeed scarce in AOF, but a more nuanced argument could be made (p.8-9), beyond this unique archive material from Dakar. Notably, I would recommend reading Barthélémy’s various works (notably on the origins of a midwife training school in 1918 in Dakar), as well as Foster and Clignet's researches.

Moreover, the lines dedicated to Rufisque should be moved from the end of p.10 to the beginning of p.9, when Second World War schooling is addressed.

In this perspective, Diawara & Aoua Keita’s marriage may be qualified as a “marriage d’évolué” following Jézéquel and Barthélémy’s analysis.

The ethnographic passage is more precise and interesting in this part. However, once again, I would recommend grounding these observations with the literature, whenever possible (for Côte d'Ivoire see for example Abou Bamba's book).

Conclusion

“In Africa, school has thus been the cause of major upheavals in the social, cultural and political order”: beyond Francophone countries then? Either delete or expand…

Smaller remarks

I am not sure to fully understand the following line: “her mother-in-law made her son - who had been hostile to polygamy until then – divorce her in 1949” (p.9), is this polygamy or divorce, or both?

Some references are missing (indicated as X).

I thank you for this rich and detailed paper, which gives a detailed overview of schooling in Francophone colonial Africa.

I look forward to reading the final version.

Author Response

I thank very much the reader for the excellent suggestions. I tried to answer in my revised text most of the remarks ; I took into account the bibliographical suggestions, even if most of them deal with the independence  period which drastically changed the educational program. What I tried to explain is the terrible slow start of colonial education, which proved, especially in FEA, unable to train westernized public officers as needed by the colonial power. This was possible only in FWA where the political will of Africans emerged much earlier, partly because of the “naturalization” of the “Originaires”. Today, the absolutely pitiful state of “modern” education in colonial and neo-colonial French-speaking Africa is difficult to imagine, yet the situation was specifically linked to the near-total absence of Africans able to gain French citizenship rights. Most Africans were “natives”, and natives had no rights. Their only hope to get French rights (Liberty, Equality, Fraternity) was to obtain French citizenship, and this was extremely unusual. It may be difficult today to understand that the main hope of most francophone Africans, as late as the 1950s, was not yet to get independence; it was to gain French citizenship. They all knew a successful example:  the exceptional French citizenship obtained in 1916 by the inhabitants borne in 4 “communes “ of Senegal : Gorée, Saint Louis, Rufisque and Dakar. It was not to be obtained elsewhere before 1956.

Round 2

Reviewer 1 Report

Due to uncertainty in the presentation about theoretical background and research methodology. Perhaps, this manuscript is best recommended as a commentary as opposed to a research study. I do not find evidence that supports the basics of qualitative or for that matter quantitative research.

Author Response

Second answer to the comment :

My paper explains facts that were never explained this way. It mostly relies on original studies, most of them never published beforehand. and explicitly commented and quoted in my bibliography. If the commentator thinks that it is a commentary rather than a comparative study between Equatorial and Westen French Africa as for their desperate results on education, I think the decision of what is my paper depends on my editors' will. I let them decide it.